# Microscopic Dynamic Mechanism of Irreversible Thermodynamic Equilibration of Crystals

Anatoly Yu. Zakharov [1,*,†] and Maxim A. Zakharov [2,†]

1    Department of General and Experimental Physics, Yaroslav-the-Wise Novgorod State University, 173003 Veliky Novgorod, Russia
2    Department of Solid State Physics, Yaroslav-the-Wise Novgorod State University, 173003 Veliky Novgorod, Russia; Maxim.Zakharov@novsu.ru
*    Correspondence: Anatoly.Zakharov@novsu.ru; Tel.: +7-905-291-2413
†    These authors contributed equally to this work.

**Abstract:** The dynamics of free and forced vibrations of a chain of particles are investigated in a harmonic model taking into account the retardation of interactions between atoms. It is found that the retardation of interactions between particles leads to the non-existence of stationary free vibrations of the crystal lattice. It is shown that in the case of a stable lattice, forced vibrations, regardless of the initial conditions, pass into a stationary regime. A non-statistical dynamic mechanism of the irreversible thermodynamic equilibration is proposed.

**Keywords:** crystal dynamics; retarded interactions; irreversibility; equilibration dynamics

## 1. Introduction

The basic principles of the dynamic theory of crystal lattices with instantaneous interactions between particles were developed, mainly, in the works of Born and their co-authors [1,2]. A further development of crystal dynamics was aimed at taking into account the features of crystal structures, models of interatomic potentials, defects in crystals, nonlinear effects, etc. [3–5]. Within the framework of this theory, a system of interacting particles is equivalent to a system of noninteracting oscillators. The dispersion law of oscillators (phonons) is related to the characteristics of interatomic potentials. It is known that interactions between particles are of field origin and, therefore, instant interactions are impossible. However, there are practically no works on the manifestation of the retardation effect of interactions in crystal dynamics.

Due to the field nature of the interaction between particles, the field is a full-fledged component of the system. Thus, the system of interacting particles itself is not closed due to the presence of an additional inevitable component—a field with an infinite set of degrees of freedom.

The study of the dynamics of few-body systems immersed in an elastic medium with an infinite set of degrees of freedom was begun in the early 20th century [6,7] (the Lamb model with an oscillator attached to an infinite string). A generalization of the Lamb model in the case of nonlinear oscillators and an inhomogeneous string was carried out in recent works [8,9]. A characteristic feature of the behavior of such systems is the damping of oscillations due to the irreversible transfer of the oscillator energy to an elastic medium. In [10–14], several few-body problems were investigated, taking into account the field retardation of interactions, and it was shown that in all the models studied there is an irreversible transfer of energy from particles to the field through which the particles interact. It should be noted that the retardation of interactions leads not only to the phenomenon of irreversibility, but also manifests itself in a qualitative change in the spectrum of elementary excitations in crystals; in particular, as shown in [15], it is responsible for the splitting of the optical branches in the dispersion curves of ionic crystals near the electromagnetic dispersion line $\omega = ck$.

It should be noted that, even in the case of a finite number of particles, the set of degrees of freedom of the field generated by them is infinite. The evolution of the system as a whole is described by equations of particles dynamics and equations of field dynamics. These equations are invariant with respect to time reversal. In particular, the complete set of solutions of equations for the potentials of the electromagnetic field contains both retarded and advanced potentials. However, the advanced potentials do not satisfy the fundamental principle of causality and, therefore, should be omitted in the study of the dynamics of systems [16]. Thus, the dynamics of a system of particles with a field origin of interactions is described by functional differential equations of the retarded type.

We emphasize that the common property of functional differential equations of the retarded type and their solutions is non-invariance with respect to time reversal, i.e., irreversibility. Note also that it is this property that radically distinguishes the laws of thermodynamics from the Newtonian laws of classical mechanics. Therefore, the study of the dynamics of systems of particles with retarded interactions between them is of interest in connection with the possible consistent microscopic substantiation of the laws of thermodynamics on a new basis. In this regard, we note that the statistical substantiation of thermodynamics is not mathematically perfect, not only because of the lack of correct resolution of the well-known paradoxes of Loschmidt and Zermelo [17–20]. Additional doubts about the correctness of the statistical approach to the microscopic substantiation of thermodynamics are raised by the Kac ring model [21,22]. This dynamic model has an exact analytical solution that is reversible in time and agrees with the Poincaré recurrence theorem. However, when very plausible assumptions such as molecular chaos are used in this model, a result is obtained that is close to the exact solution only at short times, but over time, the deviation from the exact solution increases.

This work is devoted to the study of the influence of the retardation of interatomic interactions on the dynamics of a one-dimensional crystal lattice in order to find a microscopic dynamic substantiation of the mechanism for achieving thermodynamic equilibrium in a system of particles.

## 2. Free Oscillations of a Chain with Retarded Interactions between the Particles

Consider a one-dimensional system of identical particles interacting with each other, whose stable equilibrium positions form an ideal lattice with the Born–von Karman boundary conditions [1]:

$$x_n^{(0)} = n\,a \quad \left(a = \text{const}, \quad x_{n+N}^{(0)} = x_n^{(0)}, \quad 1 \le n \le N\right), \tag{1}$$

where $a$ is the distance between the nearest neighbors of the lattice, and $N$ is the total number of particles in the system.

The local value of the potential of the field created by all particles at the site $x_n^{(0)}$ for instantaneous interactions has the form:

$$\varphi\left(x_n^{(0)}\right) = \sum_{\substack{n' \\ (n' \ne n)}} v\left(x_n^{(0)} - x_{n'}^{(0)}\right), \tag{2}$$

where $v(x)$ is the energy of a pair interaction of two atoms located at a distance $x$ from each other.

In this case, the dynamics of the system in the harmonic approximation are described by Equation [2]:

$$m\ddot{U}_n(t) = \sum_{n'>0} v''(n')\left[U_{n-n'}(t) - 2U_n(t) + U_{n+n'}(t)\right], \tag{3}$$

where $m$ is the mass of the atom, $v''(n)$ is the second derivative of the function $v(x)$ at $x = na$, and $U_n(t)$ is the displacement of the $n$-th particle from its equilibrium position:

$$U_n(t) = x_n(t) - x_n^{(0)}, \quad |U_n(t)| \ll a. \tag{4}$$

It is known that solutions of the equations of crystal lattice dynamics with instantaneous interactions in the harmonic approximation lead to the concept of phonons. However, real interactions between particles always have the property of retardation due to the finite speed of propagation of interactions. This property leads to a radical change in the dynamics, even in the simplest case of a two-body problem, including the irreversible behavior of the system [10,14].

To take into account the effect of the retardation of interactions between particles of a one-dimensional lattice in Equation (3), we determined the replacement:

$$U_{n\pm n'}(t) \longrightarrow U_{n\pm n'}(t - \tau(n'a)), \tag{5}$$

where $\tau(n'a)$ is the retardation time of interaction between points located at a distance $n'a$ from each other.

By virtue of Condition (4), we assumed that the retardation of interactions between each pair of particles depended only on the equilibrium distances between them. Since the retardation of the interaction between points was proportional to the distance between them, we determined:

$$\tau(n'a) = \frac{an'}{c} = \tau_1 \, n', \tag{6}$$

where $c$ is the speed of the propagation of interactions between particles, that is, the speed of light, and $\tau_1$ is the retardation time of the interaction between the nearest neighbors of the lattice.

Thus, the equations of the dynamics of a one-dimensional chain of interacting particles in the harmonic model, taking into account the retardation of interactions, had the following form:

$$\begin{cases} m\ddot{U}_n(t) = \sum_{n'>0} v''(n') \left[ U_{n-n'}(t - n'\tau_1) - 2U_n(t) + U_{n+n'}(t - n'\tau_1) \right]; \\ U_{n+N}(t) = U_n(t). \end{cases} \tag{7}$$

We sought a solution to this system of equations in the form:

$$U_n(t) = Q_k(t) \, e^{ikna}, \tag{8}$$

where $Q_k(t)$ are normal coordinates. It followed from the Born–von Karman boundary conditions that:

$$k = 2\pi \frac{s}{aN} \tag{9}$$

($s$ is an arbitrary integer) and:

$$-\frac{\pi}{a} \le k < \frac{\pi}{a}. \tag{10}$$

Substituting (8) into Equation (7), we obtained:

$$m\ddot{Q}_k(t) - \sum_{n'>0} v''(n') \left[ Q_k(t - n'\tau_1) \, e^{-ikan'} - 2\,Q_k(t) + Q_k(t - n'\tau_1) \, e^{ikan'} \right] = 0. \tag{11}$$

We substituted:

$$Q_k(t) = q_k \, e^{-i\omega t} \tag{12}$$

and obtained an equation for the dispersion law $\omega(\tau_1, k)$:

$$m\omega^2(\tau_1, k) - 2\sum_{n'>0} v''(n')\left[1 - e^{i\omega(\tau_1,k)\tau_1 n'}\cos(kan')\right] = 0. \tag{13}$$

This equation in the general case (that is, at $\tau_1 \neq 0$) is transcendental, and the set of its roots is infinite.

For the existence of stationary oscillations in the system, it was necessary that Equation (13) had at least one real root. We showed that for $\tau_1 \neq 0$, this equation has no real roots. We determined:

$$\omega(\tau_1, k) = \Omega(\tau_1, k) - i\Gamma(\tau_1, k) \tag{14}$$

and reduced Equation (13) with respect to the complex unknown $\omega(\tau_1, k)$ to the system of equations with respect to two real unknowns, $\Omega(\tau_1, k)$ and $\Gamma(\tau_1, k)$:

$$\begin{cases} m\left[\Omega^2(\tau_1, k) - \Gamma^2(\tau_1, k)\right] \\ -2\sum_{n'>0} v''(n')\left[1 - e^{\Gamma(\tau_1,k)\tau_1 n'}\cos(kan')\cos(\Omega(\tau_1, k)\tau_1 n')\right] = 0; \\ m\,\Omega(\tau_1, k)\Gamma(\tau_1, k) \\ + \sum_{n'>0} v''(n')e^{\Gamma(\tau_1,k)\tau_1 n'}\cos(kan')\sin(\Omega(\tau_1, k)\tau_1 n') = 0. \end{cases} \tag{15}$$

An elementary analysis showed that for $\tau_1 \neq 0$, the function $\Gamma(\tau_1, k)$ satisfies the condition:

$$\Gamma(\tau_1, k) \not\equiv 0. \tag{16}$$

Therefore, the retardation of interactions led to the impossibility of stationary free oscillations of a one-dimensional lattice. Since the retardation effect was unavoidable, the only two scenarios were possible in the harmonic approximation of the lattice dynamics.

1.    If:

$$\Gamma(\tau_1, k) > 0 \tag{17}$$

for all values $k$, then all oscillations in the system were damped and at $t \to \infty$ the oscillations ceased.

2.    If there were such values of $k$ for which:

$$\Gamma(\tau_1, k) < 0, \tag{18}$$

then the lattice would disintegrate.

We enumerated the roots of the characteristic Equation (13) under the condition $\tau_1 \neq 0$:

$$\omega_s(\tau_1, k) = \Omega_s(\tau_1, k) - i\Gamma_s(\tau_1, k), \quad s = 1, 2, \ldots. \tag{19}$$

To preserve the integrity of the lattice, it was necessary that Condition (17) was true for all $\Gamma_s(\tau_1, k)$:

$$\Gamma_s(\tau_1, k) > 0, \tag{20}$$

therefore, only this case was of interest.

Each of the roots $\omega_s(\tau_1, k)$ corresponded to an equation of free vibrations of the form:

$$\ddot{Q}_k^{(s)}(t) + 2\Gamma_s(\tau_1, k)\dot{Q}_k^{(s)}(t) + \left[\Omega_s^2(\tau_1, k) + \Gamma_s^2(\tau_1, k)\right]Q_k^{(s)}(t) = 0. \tag{21}$$

This form of the equations was used to study forced lattice vibrations.

### 3. Dynamics of Forced Oscillations of a Chain with Retarded Interactions

Consider the problem of the dynamics of a one-dimensional atomic chain immersed in an alternating external force field. We denoted the external force acting on the normal coordinate $Q_k^{(s)}(t)$ by $f_k^{(s)}(t)$. Then, the equations of the dynamics of the system had the form:

$$\ddot{Q}_k^{(s)}(t) + 2\Gamma_s(\tau_1, k)\,\dot{Q}_k^{(s)}(t) + \left[\Omega_s^2(\tau_1, k) + \Gamma_s^2(\tau_1, k)\right]Q_k^{(s)}(t) = \frac{f_k^{(s)}(t)}{m}. \tag{22}$$

Of interest was the case of (20), when the free vibrations of the atoms in the chain were damped, that is, the lattice did not disintegrate.

Because of the term $\dot{Q}_k^{(s)}(t)$, this equation had the same form as the equations of forced oscillations under friction. However, in contrast to the phenomenological approaches, here, the damping of oscillations had a microscopic, purely dynamic origin due to the finite rate of the transfer of interactions.

The general solution of Equation (22) is the sum of the general solution of the corresponding homogeneous system of Equations (which, as shown in the previous section, tends to zero as $t \to \infty$) and the particular solution of the inhomogeneous system equations. Therefore, over time, stationary oscillations are established in the system, determined by the characteristics of the external field.

We represented the external force in the form of the Fourier expansion:

$$f_k^{(s)}(t) = \sum_{s'} C_k^{(ss')}\, e^{i\widetilde{\Omega}_{s'}t}. \tag{23}$$

Then, the solution of Equation (22) had the form [23]:

$$Q_k^{(s)}(t) = \sum_{s'} \frac{C_k^{(ss')}\, e^{i\widetilde{\Omega}_{s'}t + i\delta_s(\tau_1, k)}}{m\sqrt{\left[\Omega_s^2(\tau_1, k) + \Gamma_s^2(\tau_1, k) - \widetilde{\Omega}_{s'}^2\right]^2 + 4\Gamma_s^2(\tau_1, k)\widetilde{\Omega}_{s'}^2}}, \tag{24}$$

where

$$\tan \delta_s(\tau_1, k) = \frac{2\Gamma_s(\tau_1, k)\widetilde{\Omega}_{s'}}{\widetilde{\Omega}_{s'}^2 - \Omega_s^2(\tau_1, k) - \Gamma_s^2(\tau_1, k)}. \tag{25}$$

Thus, in the limit $t \to \infty$, the system went over to a stationary state, which was in a dynamic equilibrium with an external field.

### 4. Results

The main results of this work were as follows.

1. It was found that the retardation of interactions between particles leads to a radical restructuring of the dynamics of a one-dimensional harmonic chain. In particular, due to the retardation of interactions, stationary free oscillations in the chain are impossible.
2. Since the presence of free oscillations with increasing amplitudes means the destruction of the chain, a criterion for the absence of growing oscillations in the system was obtained. This criterion is a condition for the stability of the chain.
3. It was shown that, when a stable chain of particles with retarded interactions between them is immersed in an alternating external field, the system passes into a stationary state, which depends both on the properties of the system and on the characteristics of the external field. This stationary state was interpreted as a dynamic equilibrium between a chain and an external field.

Thus, within the framework of the dynamics of a one-dimensional crystal lattice with retarded interactions between particles, the following phenomena took place:

- The phenomenon of irreversibility;
- The existence of a thermodynamic equilibrium.

Both of these phenomena are postulates both in phenomenological thermodynamics and in statistical mechanics as the zeroth law of thermodynamics. The results of this work showed that the zeroth law of thermodynamics could be substantiated, explained and described on the basis of two fundamental physical principles:

- The field nature of the interaction between particles;
- The principle of causality.

## 5. Discussion

Within the framework of pre-relativistic physics, the essence of the potential energy of interactions between particles remained a kind of "thing in itself". This is some function that depends on the instantaneous configuration of the system and has a hidden origin. In this regard, it is appropriate to note one of the first attempts to find a mechanical interpretation of the interaction of distant bodies: in the outstanding treatise [24], Heinrich Hertz proved that the potential energy of interacting bodies is mathematically equivalent to the kinetic energy of hidden particles.

In the framework of relativistic physics, an instantaneous interaction of particles distant from each other is impossible. The potential energy of a system of interacting particles, depending on their instantaneous positions, does not exist [25,26]. Therefore, a description of the dynamics of a system of atoms on the basis of the Hamiltonian of a system of particles is possible only in the non-relativistic approximation. Instead of Hertz's "hidden bodies", the field acts as a mediator in particle interactions. Therefore, a correct description of the dynamics of a system of particles interacting through the field must take into account the dynamics of the field. In particular, for the classical system of point-charged particles, such a complete system of equations consists of the equations of the relativistic dynamics of particles and the Maxwell equations for the electromagnetic field.

In article [16], a classical relativistic dynamic theory of a system of point charges interacting through the electromagnetic field created by them is constructed. The relativistic dynamics of such a system are described in terms of microscopic (i.e., not averaged) distribution functions. Within the framework of this approach, an exact analytical elimination of field variables was performed and a finite closed system of differential functional equations of the retarded type with respect to microscopic distribution functions of particles was obtained. This system of equations is not invariant with respect to time reversal, since the advanced electromagnetic fields were omitted due to the principle of causality.

The phenomenon of the damping of lattice oscillations with retarded interactions is qualitatively similar to energy dissipation in classical electrodynamics (dipole radiation). However, the physical mechanisms of these phenomena are significantly different. In the case of crystal dynamics, a nonlocal effect takes place, since each particle of the lattice interacts with the field created by all other particles, taking into account the retardation. In the case of dipole radiation, on the contrary, a local effect arises due to the accelerated movement of charges in their own electromagnetic field, that is, self-action [25].

## 6. Conclusions

Thus, the behavior of the system under study in the framework of relativistic physics was radically different from the behavior of the same system in the framework of pre-relativistic physics, in which an isolated chain of atoms oscillated eternally. In relativistic physics, both irreversibility and a state of dynamic equilibrium exist. There is neither one nor the other in pre-relativistic physics.

**Author Contributions:** Conceptualization, A.Y.Z.; Supervision, A.Y.Z.; Validation, M.A.Z.; Writing—review & editing, M.A.Z. All authors have read and agreed to the published version of the manuscript.

**Funding:** This research received no external funding.

**Acknowledgments:** We are grateful to Ya.I. Granovsky, V.V. Uchaikin, and V.V. Zubkov for the stimulating discussions. We are also sincerely grateful to A. Lerose for drawing our attention to the paper [15]. We are also grateful to the anonymous referee for interesting constructive comments that helped improve the manuscript.

**Conflicts of Interest:** The authors declare no conflict of interest.

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
