# Peer review of "Microscopic Dynamic Mechanism of Irreversible Thermodynamic Equilibration of Crystals"

_quantumrep, doi:10.3390/quantum3040045_

Round 1
Reviewer 1 Report
The manuscript by Zakharov and Zakharov titled "Microscopic dynamic mechanism of irreversible thermodynamic equilibration of crystals" investigated the free and forced vibrations of particles in lattice under non-instantaneous interaction. The topic is interesting, and the results are mainly presented clearly.
I don't have much technical comment on this manuscript. I want the authors to answer whether the following intuitions are correct:
(1) can I understand that the retardation causes dissipation into the vibration so that the particles cannot stably freely oscillate? If so, could this be an analog to the atom-light interaction where the dipole force not only serves as a conservative potential but also a dissipation force?
(2) is the growing amplitude oscillation physical? Would that be similar to the unphysical solution of dipole radiation?
I want to see the answer to these two questions before I make my final recommendation; however, overall, this is a reasonable manuscript and deserves publication.
Author Response
Response 1:
Thank you for interesting question. We think that you are right: the retardation of interactions between particles leads to the impossibility of stationary free oscillations in the harmonic model. Qualitatively, this phenomenon is analogous to the atom-light interaction, leading to dissipation, but the physical mechanisms are significantly different. In the case of dipole radiation, a local effect arises due to the accelerated motion of charges in their own electromagnetic field, i.e., self-action. In the case of crystal dynamics, a nonlocal effect takes place, since each particle of the lattice interacts with the field created by all other particles, considering the retardation.
Response 2:
Thank you for interesting question. The existence of free oscillation of the lattice with increasing amplitudes in the case of an arbitrary interparticle potential is possible. But, increasing amplitudes correspond to the situation when the lattice is unstable and disintegrates. Therefore, within the framework of the harmonic model, growing solutions are of no interest and are nonphysical. The nonphysical solution of dipole radiation is associated with the known fundamental restrictions on the applicability of classical electrodynamics: “… we must keep in mind that the description of the action of the charge "on itself" with the aid of the damping force is unsatisfactory in general and contains contradictions” (L. D. Landau, E. M. Lifshitz. The Classical Theory of Fields. Amsterdam: Butterworth Heinemann, Netherlands, 1994. P.224).
Reviewer 2 Report
The paper is interesting and clear. However, there are results in quantum mechanics suggesting that there might be some non-local interactions. It would be nice to consider this relation to your complete dismissal of advanced interactions.
Microscopic dynamic mechanism of irreversible
thermodynamic equilibration of crystals
By
- A. Yu. Zakharov and M. A. Zakharov
The authors study the influence of retardation in interatomic interactions on the dynamics of a one-dimensional crystal lattice as a means of obtaining a microscopic dynamic justification for thermodynamic equilibrium. The authors are among the experts in the field and provide a well-documented history of the important approaches, contributions, and problems. They correctly (in my opinion) object to the well-known statistical approaches and provide a clear justification for their position.
Their study is based on the design of a simple model using a finite ring of particles that are allowed to interact via delayed or retarded action. They show that, retarded interaction makes it impossible for the chain to have stationary free oscillations. This implies that the chain is a stable system. They then show that, when an alternating external field is applied, the chain passes into a stationary state, which depends on the properties of the chain and of field. They justifiably conclude that, within their model, retarded interaction induces:
- Irreversibility (non-invariance under time reversal).
- The system obtains of thermodynamic equilibrium.
Since the above results are postulates for the zeroth law of thermodynamics and statistical mechanics, the major conclusion of the authors is that the zeroth law can be explained on basis of two physical facts:
- All electromagnetic interactions are carried by the field.
- All interactions between particles are delayed (the principle of causality).
The paper is well written and technically correct. The results represent an important contribution and should be accepted for publication.
Author Response
Response:
Thank you for your interesting and helpful comments.